# Causal Visual-semantic Correlation for Zero-shot Learning

## ABSTRACT

Zero-Shot learning (ZSL) correlates visual samples and shared semantic information to transfer knowledge from seen classes to unseen classes. Existing methods typically establish visual-semantic correlation by aligning visual and semantic features, which are extracted from visual samples and semantic information, respectively. However, instance-level images, owing to singular observation perspectives and diverse individuals, cannot exactly match the comprehensive semantic information defined at the class level. Direct feature alignment imposes correlation between mismatched vision and semantics, resulting in spurious visual-semantic correlation. To address this, we propose a novel method termed **C**ausal **V**isual-**s**emantic **C**orrelation (CVsC) to learn substantive visual-semantic correlation for ZSL. Specifically, we utilize a Visual Semantic Attention module to facilitate interaction between vision and semantics, thereby identifying attribute-related visual features. Furthermore, we design a Conditional Correlation Loss to properly utilize semantic information as supervision for establishing visual-semantic correlation. Moreover, we introduce counterfactual intervention applied to attribute-related visual features, and maximize their impact on semantic and target predictions to enhance substantive visual-semantic correlation. Extensive experiments conducted on three benchmark datasets (i.e., CUB, SUN, and AWA2) demonstrate that our CVSC outperforms existing state-of-the-art methods.

## CCS CONCEPTS

• **Computing methodologies → Artificial intelligence**; **Computer vision**.

## KEYWORDS

Zero-Shot Learning, Image Classification, Visual-semantic Correlation, Causal Inference.

## 1 INTRODUCTION

Zero-Shot Learning (ZSL) stands as a significant research area in machine learning, which imitates human cognitive patterns to endow computers with ability to recognize new classes [20, 21, 30].[1] Humans can identify new classes by leveraging prior knowledge, even without direct exposure. ZSL draws inspiration from human cognition, achieving knowledge transfer from seen to unseen classes by correlating visual samples with shared semantic information

---

[1]It is noted that ZSL is typically denoted as zero-shot image classification, and we follow the standard in this paper.

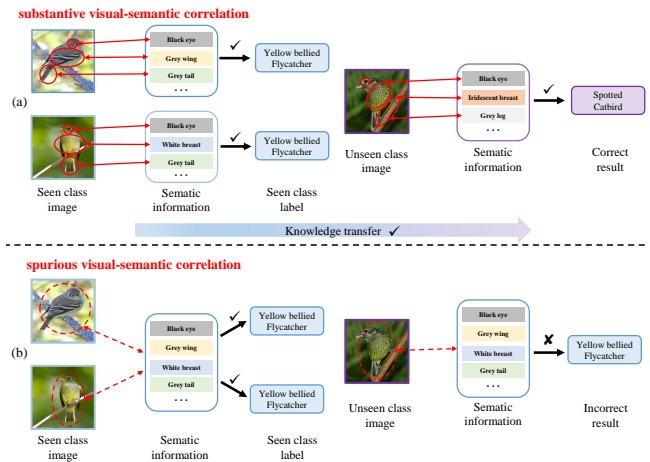

**Figure 1: Problem Analysis. (a) The ZSL model aims to establish substantive visual-semantic correlation within seen classes, facilitating accurate identification of inherent attributes in images. This enables accurate zero-shot predictions when encountering images from unseen classes by pricely identifying the genuine attributes. (b) Existing ZSL methods inevitably falls into spurious visual-semantic correlation, primarily caused by mismatches between images and their corresponding attribute annotations. Such discrepancy is due to the fact that diverse images within same class do not have the same attributes consistent with complete attribute annotations. Thus, the spurious visual-semantic correlation finally results in poor knowledge transfer.**

within seen class. Consequently, by solely relying on the semantic information of unseen classes, models can directly recognize samples from unseen classes. The semantic information typically includes attribute annotations [15, 16], word vectors [23, 28], document vectors [22, 39] and so on. In this paper, we focus on attribute annotations as the shared semantic information.

Existing methods establish the visual-semantic correlation by aligning visual and semantic features, which are extracted from instance-level images and class-level attribute annotations. Specifically, these methods map visual and semantic features into common space and design various objective functions (such as mean squared loss [52], cross-entropy loss [18], contrastive loss [17], etc.) to facilitate alignment between the two. This process ensures that images and their corresponding attribute annotations get similar representations in the common space, thus achieving visual-semantic correlation. Based on the mapping direction between visual and semantic space, existing methods are broadly divided into two categories: visual-to-semantic mapping methods [5, 25, 40, 54] and semantic-to-visual mapping methods [8, 47, 49]. The existing methods overcome the heterogeneity between vision and semantics, correlate the two and achieve knowledge transfer.

However, the mismatch between vision and semantics leads to spurious visual-semantic correlation, limiting knowledge transfer, which has not been well addressed by existing works. As shown in Figure 1(a), the model aims to establish substantive visual-semantic correlations within seen classes, enabling the identification of inherent attributes present in the images. This facilitates correct zero-shot predictions when encountering unseen class images by accurately identifying the attributes genuinely present. However, attribute annotations are typically comprehensive and defined at the class level, whereas instance-level images within same class exhibit different attributes due to varying individuals and observation perspectives. Consequently, vision and semantics cannot exactly match in practice. As illustrated in Figure 1(b), the model easily falls into spurious visual-semantic correlation due to directly aligning mismatched vision and semantics. Such spurious correlation limits knowledge transfer and results in inferior predictive performance.

Substantive visual-semantic correlation is crucial for ZSL. Inspired by the analysis above, we are prompted to utilize attribute annotations effectively to establish visual-semantic correlation, while enhancing the substantiveness of the correlation. Considering that images often contain fewer attributes than those annotated, we significantly penalize the model for predicting attributes beyond the annotations. At the same time, the model incurs minor penalties for missing attributes that present in the annotations but absent from the images, thus reasonably utilizing the attribute annotations. Furthermore, causal inference [34, 35], as an effective methodology, excels at revealing the causal correlation between variables. Hence, we leverage the tool of causal inference, specifically counterfactual intervention, to assist the model in establishing substantive visual-semantic correlation. By applying counterfactual interventions to intermediate variable within the model's visual-semantic interaction and maximizing its impact, we can enhance the substantiveness of visual-semantic correlation.

We propose Causal Visual-semantic Correlation (CVsC) to establish a substantive visual-semantic correlation for ZSL. Specifically, CVsC utilizes the Visual Semantic Attention module to facilitate interaction between vision and semantics, thereby identifying attribute-related visual features. Subsequently, a Semantic Embedding module maps these features to obtain semantic vector and determine the category. Furthermore, we design Conditional Correlation Loss to effectively utilize semantic information as supervision. Finally, we introduce counterfactual causal intervention applied to attribute-related visual features, and maximize their impact on semantic and target predictions to enhance substantive visual-semantic correlation. The comprehensive experimental demonstrates that CVsC effectively mitigates spurious visual-semantic correlation and significantly enhances the performance of ZSL.

In summary, our contributions are as follows:

- We explore the significance of visual-semantic correlation for ZSL and highlight that direct visual-semantic alignment leads to spurious visual-semantic correlation, thereby constraining knowledge transfer.
- We design Causal Visual-semantic Correlation (CVsC) to establish substantive visual-semantic correlation.

- Extensive experiments on multiple benchmark datasets demonstrate that CVsC enhances the substantiveness of visual-semantic correlation and significantly improves the performance of ZSL.

## 2 RELATED WORKS

### 2.1 Zero-Shot Learning

Zero-Shot Learning (ZSL) [20, 30], which is proposed to address the data dependency issue in machine learning, can directly identify unseen class samples. Based on the classes included in the testing phase, ZSL is typically divided into conventional settings and generalized settings [37]. In conventional ZSL (CZSL), testing samples only contain unseen classes, while in generalized ZSL (GZSL), testing samples come from both seen and unseen classes. Inspired by human cognition, ZSL transfers knowledge from seen to unseen classes by correlating visual samples with corresponding semantic information within seen classes [9, 42, 47, 48]. The correlation between vision and semantics is established by aligning visual and semantic features, which are extracted from instance-level images and class-level semantic information. Existing methods typically map images and semantic information into a common space and then design objective functions to achieve feature alignment. Based on the mapping direction, existing methods are mainly classified into two categories: visual-to-semantic mapping methods [3, 5, 25, 40, 54] and semantic-to-visual mapping methods [8, 42, 47, 49, 57]. These methods overcome the heterogeneity and correlate visual and semantic information, thereby achieving knowledge transfer.

Despite significant efforts in this field, the challenge of spurious visual-semantic correlation continues to hamper ZSL performance. Since instance-level images cannot strictly match attribute annotations defined at the class level, the models easily fall into spurious visual-semantic correlation. To address this problem, we propose CVsC, which enhances the substantiveness of visual-semantic correlation by effectively utilizing attribute annotations and introducing causal inference to correlate vision and semantics.

### 2.2 Attention Mechanism in ZSL

In ZSL, refining the association between visual and semantic features by aligning local visual features can lead to improved performance [6, 18]. Early explorations of local representations in ZSL relied on part detection methods [14, 50], which utilize pre-trained part detectors for local region representation. However, such methods are limited by the additional and costly annotation data required by the part detectors. Attention mechanism, benefiting from its expertise in extracting discriminative local features, has been introduced into ZSL. [53] initially proposed a stacked semantic-guided attention method, enabling the model to learn more discriminative local visual features. [58] introduced a semantic-guided multi-attention localization model, which can identify the most discriminative parts of objects without additional supervisory guidance. [18] decoupled class semantic vectors into multiple attribute vectors, and determined attribute-related visual features by computing attention maps between attribute vectors and local visual features. In this paper, we adopt the visual-semantic interaction approach from [18] to design our model.

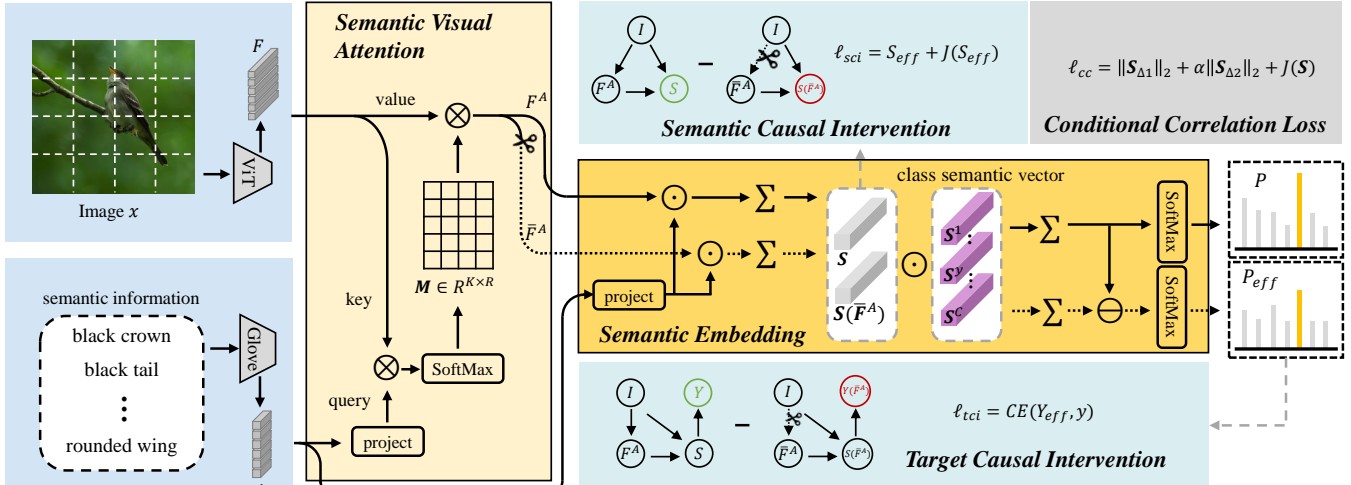

**Figure 2: The framework of Causal Visual-semantic Correlation (CVsC). CVsC first takes ViT [13] and Glove [36] to extract regional visual features and attribute vectors, respectively. Next, the Semantic-Visual Attention module facilitates interaction between regional visual features and attribute vectors, identifying attribute-related visual features. Finally, the Semantic Embedding module maps semantic vectors, enabling computation of probabilities for unseen classes. To establish substantive visual-semantic correlation, CVsC employs Conditional Correlation Loss to properly utilize attribute annotations. Additionally, CVsC introduces causal interference, i.e. Semantic Causal Interference and Target Causal Interference, to strengthen the substantive correlation between vision and semantics.**

## 2.3 Causal Inference

Causal inference [32, 34, 35, 41] investigates the effects of variables when some cause is changed, providing researchers with an effective tool to determine the substantive correlation between variables. In recent years, there has been a rapid growth of interest in combining deep learning with causal inference. It has been successfully applied in various fields, including explainable machine learning [29], natural language processing [45, 51], computer vision [27, 38, 43], and others. The substantive visual-semantic correlation is crucial for ZSL. Therefore, in this paper, we introduce causal inference to ZSL to establish a substantive visual-semantic correlation. By imposing counterfactual interventions on attribute-related visual features learned by the model, we can observe their impact on the model. By maximizing the impact of counterfactual intervention, we enhance the effectiveness of attribute-related visual features, thereby establishing substantive visual-semantic correlation.

## 3 METHODS

In this section, we first revisit ZSL setting. Then, we introduce the detailed design of Causal Visual-semantic Correlation (CVsC), the flowchart of which is illustrated in Figure 2.

**ZSL setting** ZSL has two sets of classes, i.e. seen classes $C^s$ and unseen classes $C^u$. The corresponding samples are denoted as $x \in \mathcal{X}^s$ for seen classes and $x \in \mathcal{X}^u$ for unseen classes, with labels $y \in \mathcal{Y}^s$ for seen classes and $y \in \mathcal{Y}^u$ for unseen classes. The class-level attribute annotations are encoded to semantic vectors. Here, we denote the semantic vector of class $c$ as $S^c = \left[s_1^c, \ldots, s_K^c\right]^\top$, where $s_c^k$ represents the value of the $k$-th attribute in class $c$, and $K$ represents the total number of attributes. The class semantic vectors are available for all classes in both the training and testing phases. The training set $\mathcal{D}^{train} = \{(x, y)\}$ consists of samples from seen

classes and their corresponding labels. ZSL is categorized into two types: conventional ZSL (CZSL) and generalized ZSL (GZSL) based on the scope of the testing set. The goal of CZSL is to predict image labels from unseen classes ($x \in \mathcal{X}^u$) in the testing set $\mathcal{D}^{test} = \{x\}$, while GZSL aims to predict image labels from both seen and unseen classes($x \in \mathcal{X}^s \cup \mathcal{X}^u$ in the testing set $\mathcal{D}^{test} = \{x\}$).

CVsC takes image and attribute as the input, and employs Semantic Visual Attention to interact with vision and semantics, thereby obtaining attribute-related visual features. Subsequently, a Semantic Embedding module maps these features into the semantic space, and calculates the similarity with the class semantic vectors to obtain the class probabilities. In order to achieve substantive visual-semantic correlation, CVsC first employs Conditional Correlation Loss to reasonably utilize semantic information as supervision. Furthermore, we introduce causal interference into CVsC, i.e. Semantic Causal Interference and Target Causal Interference. By imposing counterfactual interventions on attribute-related visual features learned by the model, and maxing their impact, we can get more effective features and strengthen the substantive correlation between vision and semantics.

## 3.1 Semantic Visual Attention

As shown in Figure 2, the Semantic Visual Attention takes the regional visual features and attribute vectors as input. In particular, The image $x$ is divided into $R$ patches, and features corresponding to these patches are extracted using the Vision Transformer (ViT) [13]. Here, we utilize the output of the last layer of ViT, removing the cls token, as the regional visual features, denoted as $F = [f_1, ..., f_r, ...f_R]^T$. Simultaneously, we employ Glove [36], the pre-trained language models, to extract attribute names as attribute vectors, denoted as $A = [a_1, ..., a_k, ..., a_K]^T$.

Subsequently, we utilize attention mechanisms to facilitate interaction between vision and semantics. Here, we use $Q$, $K$, and $V$ to respectively denote query, key, and value. They are defined as follows:

$$Q = AW_q, K = F, V = F. \tag{1}$$

By taking the attribute as the query, we compute its correlation matrix with the key (i.e., regional visual feature $F$) to obtain the attribute attention map $M$, and scale it using SoftMax. Here, the attribute vectors $A$ are projected to ensure uniform dimensions for $Q$, $K$, and $V$, ensuring their dot product computation. Then, the attribute-related visual features $F^A$ are obtained by multiplying the attribute attention map $M$ with the $V$:

$$F^A = MV = \text{softmax}(QK)V. \tag{2}$$

## 3.2 Semantic Embedding

Next, we employ the Semantic Embedding module to map attribute-related visual features $F^A$ to the semantic space, obtaining the semantic vector for the image, denoted as $S = [s_1, \ldots, s_K]^\top$. Subsequently, the final class probabilities are obtained by computing the similarity between this embedding semantic vector and the class semantic vectors.

Here, Semantic Embedding module project the attribute vectors $A$ to unify the dimensions with $F^A$, and compute their similarity to obtain the attribute score $s_k$, given by:

$$s_k = (a_k W_{sv}) f_k^{A\top}. \tag{3}$$

After obtaining the semantic vector $S$, class probabilities with scaling can be obtained by calculating the similarity between the mapping semantic vector and the class semantic vector, which can be formulated as:

$$p^c = \text{softmax}\left(S^\top S^c\right)$$
$$= \frac{\exp\left(\sum_{k=1}^K s_k \times s_k^c\right)}{\sum_{c' \in C} \exp\left(\sum_{k=1}^K s_k \times s_k^{c'}\right)}. \tag{4}$$

To optimize the model, existing methods typically take cross entropy loss and calibration loss [6, 18, 26], which can be expressed as:

$$\mathcal{L} = \mathcal{L}_{ce} + \lambda_{cal}\mathcal{L}_{cal}$$
$$= -\log\left(p^y\right) - \lambda_{cal}\log\left(\sum_{c \in C_u} p^c\right), \tag{5}$$

the $\lambda_{cal}$ is the weight to control the weight coefficient of calibration loss.

We can find that such objective functions primarily utilize class labels as supervision, while ignoring establishing substantive visual-semantic correlation for ZSL. Therefore, building upon this, we design Conditional Correlation Loss to effectively leverage semantic information, and introduce counterfactual causal intervention (including Semantic Causal Intervention and Target Causal Intervention) to strengthen the substantive visual-semantic correlation.

## 3.3 Conditional Correlation Loss

As analyzed in the introduction, the mismatch between images and corresponding attribute annotations hinders substantive visual-semantic correlation. Such mismatch is caused because instance-level images can not match the complete attribute annotations defined at the class level.

Given that the attributes present in images are typically less than attribute annotations, we impose major penalties on the model for predicting attributes beyond the annotations. Simultaneously, minor penalties are applied to the model for failing to predict attributes present in the annotations but absent from the images, thus facilitating effective utilization of attribute annotations. Therefore, we define two semantic difference vectors:

$$\Delta S_1 = \begin{bmatrix} max(s_1 - s_1^y, 0) \\ \vdots \\ max(s_K - s_K^y, 0) \end{bmatrix}, \Delta S_2 = \begin{bmatrix} max(s_1^y - s_1, 0) \\ \vdots \\ max(s_K^y - s_K, 0) \end{bmatrix}. \tag{6}$$

$\Delta S_1$ represents the excess of predicted semantic information in the images beyond the predefined semantic information, and $\Delta S_2$ represents the opposite. By imposing differentiated penalties on $\Delta S_1$ and $\Delta S_2$, we can effectively utilize semantic information. Meanwhile, we introduce a regularization term $J(S)$ to constrain the distribution of $S$, which maintains the standard deviation of $S$ within the same range as $S^y$. Therefore, the Conditional Correlation Loss is formulated as:

$$\mathcal{L}_{cc} = \|\Delta S_1\|_2 + \alpha \|\Delta S_2\|_2 + J(S)$$
$$= \|\Delta S_1\|_2 + \alpha \|\Delta S_2\|_2 + \left\|\text{var}\left(S^y\right) - \text{var}(S)\right\|_2. \tag{7}$$

Here, $\alpha$ is the coefficient to control the differentiated penalty weight, we set it to 0.5.

## 3.4 Causal Intervention for ZSL

Here, we first present the causal perspective within the CVsC, followed by the introduction of Semantic Causal Intervention and Target Causal Intervention that we designed.

**Causal View of ZSL** We introduce the formulation of causality for CVsC by using causal graph (also known as structural causal model [33]). The causal graph is a directed acyclic graph $\mathcal{G} = \{\mathcal{N}, \mathcal{E}\}$, where each variable in the model corresponds to a node in $\mathcal{N}$, and the causal links in $\mathcal{E}$ describe how these variables interact. As depicted in Figure 3(a), we utilize nodes in the causal graph to represent the variables involved in our model, including the model input $I$ (comprising images and attributes), $F^A$ (which corresponds to attribute-related visual features), semantic prediction $S$, and target prediction $Y$. The link $I \rightarrow F^A$ denotes the model uses Semantic Visual Attention to determine the variable $F^A$. The link $(I, F^A) \rightarrow S$ indicates that the model combines $I$ and $F^A$ to predict

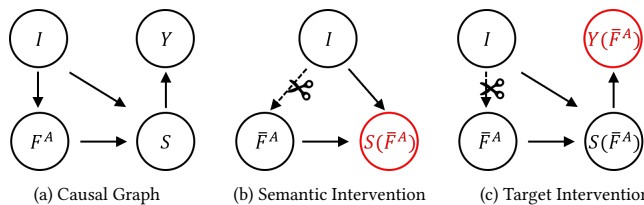

(a) Causal Graph     (b) Semantic Intervention     (c) Target Intervention

**Figure 3: The causal graphs of ZSL.**

$S$. As for $S \to Y$, it signifies the prediction of image categories based on $S$.

The variable $F^A$, as the product of visual-semantic interaction, reflects and influences the substantive nature of visual-semantic correlation. Traditional methods primarily rely on seen class labels to optimize models and treat the model as a black box, overlooking the impact of the quality of $F^A$ on semantic and target predictions. Causal inference provides a tool to break free from the black box by analyzing causal relationships between variables, aiding us in critical analysis. Therefore, we utilize causality to assess the effectiveness of learned $F^A$, then encourage the model to learn more effective $F^A$ to establish substantive visual-semantic correlation.

The introduction of a causal graph enables us to analyze the causalities by manipulating variables and observing their effects. Such operation is termed intervention in the causal inference [34, 41] and can be represented as $do(\cdot)$. When investigating the impact of a variable, the intervention involves removing all incoming links to the variable and assigning it a new value. For instance, in our causal graph, $do(F^A = \bar{F}^A)$ denotes setting the variable $F^A$ to $\bar{F}^A$ and severing the link $I \to F^A$, thereby removing the causal influence from its parent variable $I$.

Inspired by existing causal inference methods [33, 38, 41], we incorporate counterfactual intervention to investigate the impact of $F^A$. The counterfactual intervention is achieved by an imaginary intervention altering the state of the variables assumed to be different [38, 41]. In our work, we conduct counterfactual intervention $do(F^A = \bar{F}^A, I = (F, A))$ by replacing the learned attribute-related visual features $F^A$ with non-existent features $\bar{F}^A$ while keeping the input $I$ unchanged. Subsequently, we can observe its impacts on semantic and target predictions, as depicted in Figure 3(a)(b). By maxing these impacts, we can make the model learn more efficient $F^A$. The specific operations will be detailed in subsequent sections.

*3.4.1 Semantic Causal Intervention.* The variable $F^A$ first affects the semantic prediction results. Thus, we observe its impacts on semantic prediction. By substituting the learned attribute-related visual features $F^A$ with imaginary features $\bar{F}^A$, we implement a counterfactual intervention $do(F^A = \bar{F}^A)$. In practice, we replace $F^A$ with randomly generated features of the same dimensions. Referring to [33, 38, 41], we can assess the influence of variable $F^A$ by observing the difference between original semantic predictions $S(F^A = F^A, I = (F, A))$ and semantic predictions under counterfactual intervention $S(do(F^A = \bar{F}^A), I = (F, A))$:

$$S_{eff} = S(F^A = F^A, I = (F, A)) - S(do(F^A = \bar{F}^A), I = (F, A)). \tag{8}$$

$S_{eff}$ reflects the impact of $F^A$ for the semantic prediction, thus maximizing the $S_{eff}$ can guide the model to learn more effective $F^A$. Additionally, we introduce a regularization term to prevent the model from simply weakening $S(do(F^A = \bar{F}^A), I = (F, A))$, which utilize class semantic vectors to constrain $S_{eff}$. Consequently, the Semantic Causal Intervention can be finally formulated as the following objective function:

$$\begin{aligned} \mathcal{L}_{sci} &= S_{eff} + J(S_{eff}) \\ &= \sum_{k=1}^{k=K} s_k - s_k\left(\bar{F}^A\right) + \left\| S^y - \left(S - S\left(\bar{F}^A\right)\right)\right\|_2. \end{aligned} \tag{9}$$

*3.4.2 Target Causal Intervention.* The variable $F^A$ also influences the final target prediction. Therefore, we simultaneously maximizing the impact of $F^A$ on target prediction to learn effective $F^A$. Similarly, the influence of $F^A$ on the target prediction $Y$ can be represented by the difference between the original $Y(F^A = F^A, I = (F, A))$ and the counterfactual intervention $Y(do(F^A = \bar{F}^A), I = (F, A))$:

$$Y_{eff} = Y(F^A = F^A, I = (F, A)) - Y(do(F^A = \bar{F}^A), I = (F, A)). \tag{10}$$

Here, we employ cross-entropy to design the objective function for the Target Causal Intervention, which can be expressed as:

$$\begin{aligned} \mathcal{L}_{tci} &= CE(Y_{eff}, y) \\ &= -\sum \log\left(p_{eff}^y\right). \end{aligned} \tag{11}$$

Through the collaborative effort between Semantic Causal Intervention and Target Causal Intervention, the model is encouraged to learn effective attribute-related visual features, thereby establishing substantive visual-semantic correlation.

## 3.5 Optimization and Zero-Shot Prediction

To optimize our CVSC, we need to minimize the overall objective function, which comprises the typical loss in 5, Conditional Correlation Loss, Semantic Causal Intervention and Target Causal Intervention. This can be represented as:

$$\mathcal{L}_{CVsC} = \mathcal{L} + \lambda_{cc}\mathcal{L}_{cc} + \lambda_{sci}\mathcal{L}_{sci} + \lambda_{tci}\mathcal{L}_{tci}, \tag{12}$$

where $\lambda_{cc}$, $\lambda_{sci}$ and $\lambda_{tci}$ are the weight to control the Conditional Correlation Loss, Semantic Causal Intervention, and Target Causal Intervention, respectively.

After completing model training, we can directly take the model to predict the unseen images under the CZSL setting. While, for the GZSL setting, where test images come from both seen and unseen classes, the calibration factor [4] is employed to adjust the model bias towards seen classes. Specifically, the GZSL predicted expression is defined as:

$$p_{gzsl}^c = \arg\max_{c \in C^u \cup C^s}\left(p^c + \delta\mathbb{I}_{[c \in C^u]}\right), \tag{13}$$

where $\delta_1$ and $\delta_2$ are the calibration factor corresponding to two subnets, and $\mathbb{I}_{[c \in C^u]}$ is an indicator function.

## 4 EXPERIMENTS

In this section, we introduce the datasets, evaluation protocols, and implementation details. Furthermore, we provide a series of experiment analyses to verify our method.

**Datasets.** We conduct extensive experiments on three widely used benchmark datasets, i.e., AWA2 [46], CUB [44] and SUN [31]. AWA2

**Table 1: Detailed illustration for the ZSL benchmark datasets. $s$ and $u$ represent seen and unseen classes, respectively.**

| Dataset | # images | # classes ($s|u$) | # attributes |
|---------|----------|-------------------|--------------|
| CUB [44] | 11788 | 200 (150 \|50) | 312 |
| SUN [31] | 14340 | 717 (645 \|72) | 102 |
| AWA [46] | 37322 | 50 (40 \|10) | 85 |

**Table 2: Results (%) of the state-of-the-art under CZSL and GZSL settings on AWA2, CUB and SUN. The best and the second best results are marked in red and blue, respectively. The symbol '-' indicates no results.**

| Methods | Backbone | Image size | CUB | | | | SUN | | | | AWA2 | | | |
|---------|----------|-----------|-----|---|---|---|-----|---|---|---|------|---|---|---|
| | | | CZSL | GZSL | | | CZSL | GZSL | | | CZSL | GZSL | | |
| | | | acc | U | S | H | acc | U | S | H | acc | U | S | H |
| f-VAEGAN-D2 [49] | ResNet101 | 224×224 | 61.0 | 48.4 | 60.1 | 53.6 | 64.7 | 45.1 | 38.0 | 41.3 | 71.1 | 57.6 | 70.6 | 63.5 |
| FREE [8] | ResNet101 | 224×224 | - | 55.7 | 59.9 | 57.7 | - | 47.4 | 37.2 | 41.7 | - | 60.4 | 75.4 | 67.1 |
| HSVA [9] | ResNet101 | 224×224 | 62.8 | 52.7 | 58.3 | 55.3 | 63.8 | 48.6 | 39.0 | 43.3 | - | 59.3 | 76.6 | 66.8 |
| CE-GZSL [17] | ResNet101 | 224×224 | 77.5 | 63.9 | 66.8 | 65.3 | 63.3 | 48.8 | 38.6 | 43.1 | 70.4 | 63.1 | 78.6 | 70.0 |
| DAZLE [18] | ResNet101 | 224×224 | 66.0 | 56.7 | 59.6 | 58.1 | 59.4 | 52.3 | 24.3 | 33.2 | 67.9 | 60.3 | 75.7 | 67.1 |
| HAS [11] | ResNet101 | 224×224 | 76.5 | 69.6 | 74.1 | 71.8 | 63.2 | 42.8 | 38.9 | 40.8 | 71.4 | 63.1 | 87.3 | 73.3 |
| MSDN [7] | ResNet101 | 448×448 | 76.1 | 68.7 | 67.5 | 68.1 | 65.8 | 52.2 | 34.2 | 41.3 | 70.1 | 62.0 | 74.5 | 67.7 |
| TransZero [6] | ResNet101 | 448×448 | 76.8 | 69.3 | 68.3 | 68.8 | 65.5 | 52.6 | 33.4 | 40.8 | 70.1 | 61.3 | 82.3 | 70.2 |
| IEAM-ZSL [2] | ViT-Large | 224×224 | - | 68.6 | 73.8 | 71.1 | - | 48.2 | 54.7 | 51.3 | - | 53.7 | 89.9 | 67.2 |
| ViT-ZSL [1] | ViT-Large | 224×224 | - | 67.3 | 75.2 | 71.0 | - | 44.5 | 55.3 | 49.3 | - | 51.9 | 90.0 | 68.5 |
| DUET [10] | ViT-Base | 224×224 | 72.3 | 62.9 | 72.8 | 67.5 | 64.4 | 45.7 | 45.8 | 45.8 | 69.9 | 63.7 | 84.7 | 72.7 |
| PSVMA [24] | ViT-Base | 224×224 | - | 70.1 | 77.8 | 73.8 | - | 61.7 | 45.3 | 52.3 | - | 73.6 | 77.3 | 75.4 |
| CVsC | ViT-Base | 224×224 | 79.1 | 72.4 | 78.4 | 75.3 | 71.5 | 61.9 | 47.6 | 53.8 | 73.1 | 68.0 | 87.0 | 76.4 |

contains 37,322 images from 50 animal categories with 85 attributes. CUB contains 11,788 images from 200 bird categories with 312 attributes. SUN consists of 14,340 images from 717 scene classes with 102 attributes. For each dataset, we followed the recommended splits [46], dividing the classes into seen and unseen, as detailed in Table 2.

**Evaluation Protocols.** The performance of ZSL is evaluated by testing the average top-1 accuracy for each class. In the CZSL setting, we calculate the accuracy ($Acc$) by predicting the unseen classes on the test samples. In GZSL, which testing set consists of both seen and unseen samples, we need to evaluate the accuracy separately for the seen classes ($S$) and unseen classes ($U$). Therefore, the performance of GZSL is ultimately assessed by using their harmonic mean, defined as $H = (2 \times S \times U)/(S + U)$ [46].

**Implementation Details.** We implemented our method by using the PyTorch framework[2]. We employed the ViT [13] pre-trained on ImageNet [12] as the backbone for visual feature extraction. The input images were resized to $224 \times 224$. The attribute vectors were extracted by using the GloVe model [36] trained on Wikipedia articles. The model was optimized by using the Adam [19] optimizer on an NVIDIA 3090.

### 4.1 Comparison with State-of-the-Arts

We compute the performance of CVsC under the CZSL and GZSL settings on the CUB, SUN, and AWA2. The results of CVsC are compared with state-of-the-art methods employing different visual backbones, such as TransZero [6] and MSDN [7] using ResNet as the backbone, and PSVMA [24] using ViT as the backbone. Table 2 shows the comparison results. In the CZSL setting, our CVsC achieves the best average top-1 accuracies of 79.1%, 71.5% and 73.1% on CUB, SUN and AWA2, respectively. And under the GZSL setting, CVsC also gets the best results of harmonic mean, e.g., 75.3%, 53.8% and 76.4% on CUB, SUN and AWA2, respectively. These results demonstrate the superiority of the proposed method, which

---

[2]https://pytorch.org/

**Table 3: Results (%) of CZSL and GZSL ablation study on CUB, SUN and AWA2. The CCL represents Conditional Correlation Loss, The SCI represents Semantic Causal Intervention, and TCI represents Target Causal Intervention.**

| Methods | CUB | | SUN | | AWA2 | |
|---------|-----|---|-----|---|------|---|
| | CZSL | GZSL | CZSL | GZSL | CZSL | GZSL |
| | Acc | H | Acc | H | Acc | H |
| Baseline | 74.3 | 66.9 | 68.1 | 48.7 | 59.8 | 58.9 |
| CVsC w/o CCL | 77.6 | 71.4 | 69.4 | 49.2 | 65.4 | 62.8 |
| CVsC w/o SCI | 77.6 | 73.9 | 71.2 | 53.2 | 72.5 | 76.3 |
| CVsC w/o TCI | 77.1 | 73.0 | 70.3 | 51.7 | 70.1 | 73.8 |
| CVsC | 79.1 | 75.3 | 71.5 | 53.8 | 73.1 | 76.4 |

benefits from establishing substantive visual-semantic correlations, enabling effective knowledge transfer in ZSL.

### 4.2 Ablation

To gain further insights into CVsC, we conducted ablation studies to evaluate the effectiveness of its key components, namely Conditional Correlation Loss, Semantic Causal Intervention and Target Causal Intervention.

Here, we take the model with the same model architecture but without the above three main modules as the baseline. Subsequently, we compared the baseline with CVsC models without Conditional Correlation Loss (denoted as CVsC w/o CC), without Semantic Causal Intervention (denoted as CVsC w/o SCI), and without Target Causal Intervention (denoted as CVsC w/o TCI). We conducted experiments on CUB, SUN, and AWA2 datasets, and the results are presented in Table 3. It shows that CVsC exhibits significant performance improvement relative to the baseline, while the absence of the aforementioned three components leads to performance degradation. This indicates that all three components effectively assist the model in constructing substantive visual-semantic correlation, thereby enhancing ZSL performance.

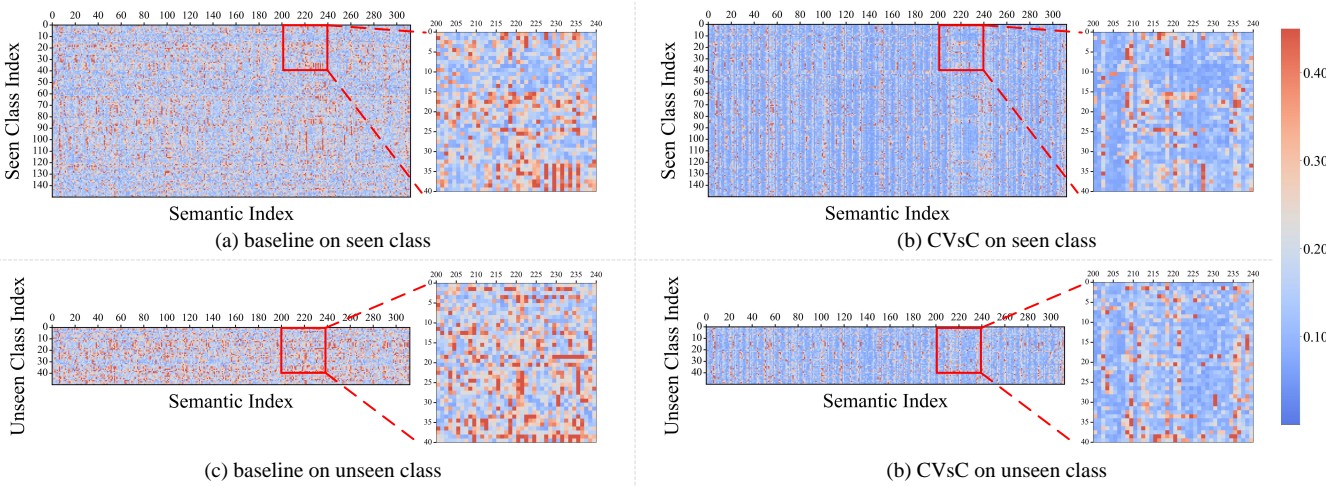

**Figure 4: Semantic error visualization on the test set of CUB. (a)(c) are the class-averaged semantic error matrix for the baseline on seen and unseen class. And (b)(d) are the class-averaged semantic error matrix for CVsC on seen and unseen class.**

## 4.3 Qualitative Evaluation for Visual-semantic Correlation

In addition to the improvement in accuracy, we conducted quantitative experiments to further demonstrate the ability of our CVSC to establish more substantive visual-semantic correlation, including semantic error visualization and attribute attention map visualization.

*4.3.1 Semantic Error Matrix.* Inspired by [52], we calculated the class-averaged semantic error by computing the difference between predicted attribute values $s_k$ and annotated values $s_k^c$ of class $c$ on the test set. We normalize the predicted attribute values in the same manner as the annotated attribute values, mitigating the influence of data outliers on the computation of class-averaged semantic error. Therefore, the class-averaged semantic error can be calculated as follows:

$$e_{ck} = \frac{1}{|\mathcal{N}_c|} \sum_{n \in \mathcal{N}_c} (s_k^n - s_k^c)^2, \tag{14}$$

where $\mathcal{N}_c$ the sample set of class $c$ on the test set, and $|\mathcal{N}_c|$ is the number of corresponding samples.

We adopt the same baseline as in the ablation experiment and compute the class-averaged semantic error for both the baseline and CVsC on CUB. Then, we visualize the class-averaged semantic errors of both the baseline and CVsC for seen and unseen classes separately using matrices, as illustrated in Figure 4. It can be observed that CVsC significantly reduces semantic errors for both seen and unseen classes.

Furthermore, we statistically analyzed the overall average errors for seen and unseen classes, and calculated the increase in error for unseen classes relative to seen classes, as shown in Table 4. It can be observed that CVsC not only has lower semantic errors compared to the baseline but also exhibits significantly lower error increases for seen classes.

**Table 4: Semantic error statistics on the test set of CUB.**

| method | seen class average error | unseen class average error |
|---|---|---|
| Baseline | 0.146 | 0.174 ↑19.1% |
| CVsC | 0.089 | 0.099 ↑11.9% |

These results demonstrate that our CVsC can enhance the substantiveness of visual-semantic correlation, facilitating knowledge transfer from seen to unseen classes.

*4.3.2 Visualization of Attribute Attention Map.* We visualized the model's attribute attention maps, implemented by computing the attribute attention map $M$ defined in Eq. 2, to further demonstrate the role of CVsC in establishing substantive visual-semantic correlation.

Here, we conducted visualizations separately for both the baseline and CVsC on CUB. The results shown in Figure 5, reveal numerous attribute localization errors in the baseline, whereas CVsC significantly improves upon this situation. This observation suggests that although ViT demonstrates strong representational capabilities, deeper layers are more susceptible to attention collapse [56], leading to challenges in attribute localization compared to convolutional networks. This difficulty impedes the model from accurately associating attributes with specific regions, resulting in spurious visual-semantic correlation and limiting knowledge transfer. However, our CVsC approach facilitates the establishment of substantive visual-semantic correlation, enabling accurate attribute localization.

## 4.4 Hyperparametric Analysis

There are three key factors involved in our method, the weights of Conditional Correlation Loss, Semantic Causal Intervention and Target Causal Intervention, i.e., $\lambda_{cc}$, $\lambda_{sci}$ and $\lambda_{tci}$. To analyze the robustness of our CVsC and select optimal hyperparameters for it, we try a wide range of $\lambda_{cc}$, $\lambda_{sci}$ and $\lambda_{tci}$ evaluated on CUB. The results are shown in Figure 6. It shows that CVsC is robust when the

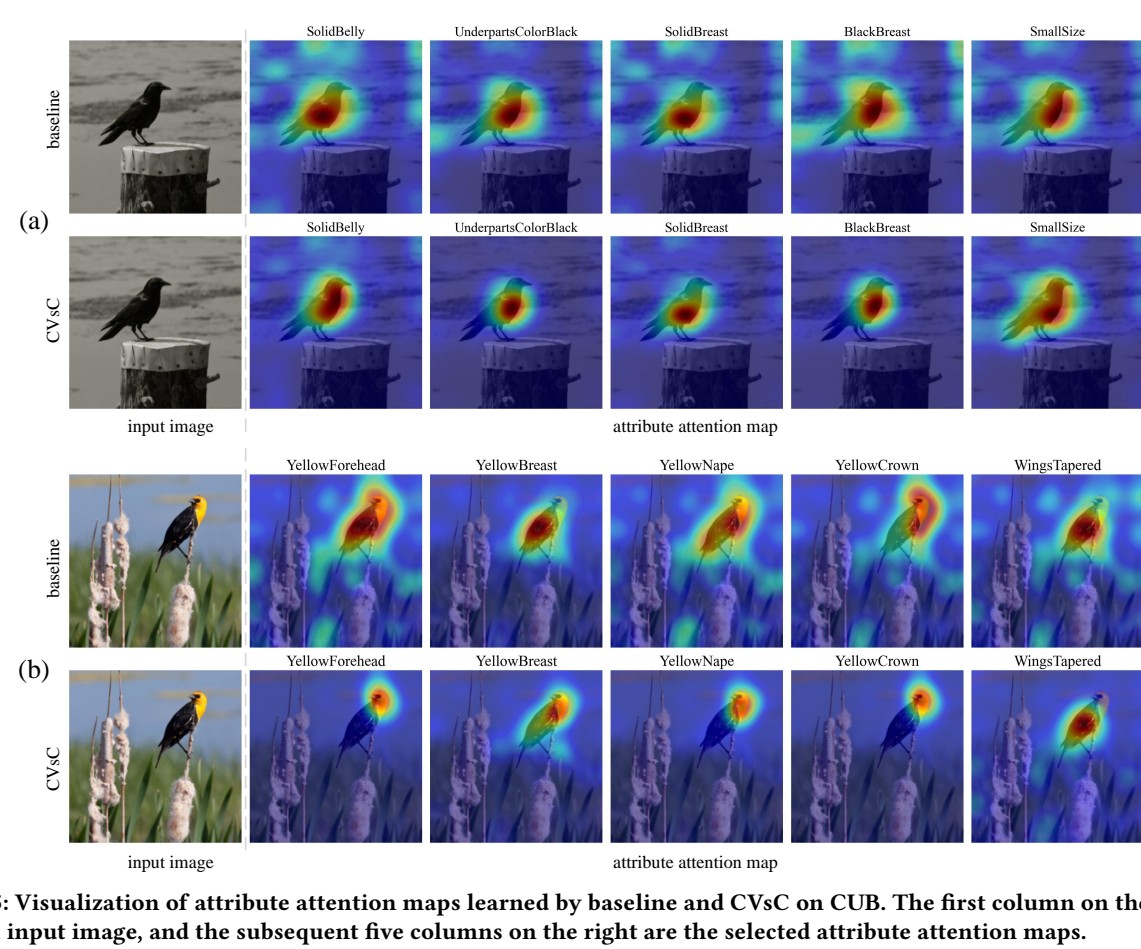

**Figure 5: Visualization of attribute attention maps learned by baseline and CVsC on CUB. The first column on the left is the selected input image, and the subsequent five columns on the right are the selected attribute attention maps.**

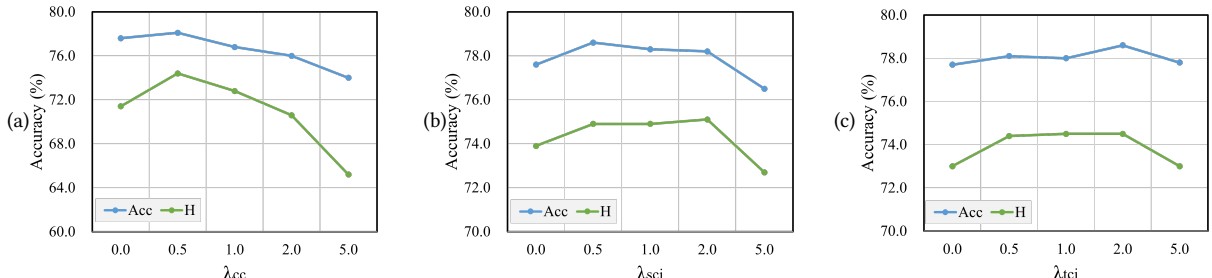

**Figure 6: Hyperparameter analysis of $\lambda_{cc}$, $\lambda_{sci}$ and $\lambda_{tci}$. We show the CZSL and GZSL performance variations on CUB.**

loss weights are set to small, while the performance drops rapidly when loss weights are set to too large. Because the large loss weights will hamper the balance of various losses. According to the results in Figure 6, we set the weights $\{\lambda_{cc}, \lambda_{sci}, \lambda_{tci}\}$ to $\{0.5, 2.0, 2.0\}$.

## 5 CONCLUSION

In this paper, we emphasize the significance of visual-semantic correlation for ZSL, and highlight the presence of spurious visual-semantic correlation caused by mismatches between instance-level images and class-level attribute annotations. This inspires us to propose a novel method termed Causal Visual Semantic Correlation (CVsC), which can effectively enhance the substantive correlation between vision and semantics. CVsC first employs Conditional Correlation Loss to properly use attribute annotations as supervision for establishing visual-semantic correlation. Furthermore, it integrates causal inference techniques to strengthen the substantive correlation between vision and semantics. Specifically, it achieves this by applying counterfactual interventions to intermediate variables learned during visual-semantic interactions, and maximizing their impact on semantic and target prediction. Extensive experiments demonstrate the effectiveness of our proposed model, achieving state-of-the-art performance by establishing substantive visual-semantic correlation.

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

Received 20 February 2007; revised 12 March 2009; accepted 5 June 2009

