# OpenReview forum: "Causal Visual-semantic Correlation for Zero-shot Learning"
_acmmm.org/ACMMM/2024/Conference — MM2024 Poster_

### Official Review · Reviewer_NqGy · 2024-05-22

**Rating:** 4
**Confidence:** 2

**Summary:**

Zero-Shot Learning bridges visual data with semantic descriptions to generalize to unseen classes, conventionally aligning visual and semantic features.
However, individual images rarely encapsulate the full class-level semantics, causing misaligned correlations.
This tackles this by employing Visual Semantic Attention for attribute-focused feature extraction, Conditional Correlation Loss for guided alignment, and counterfactual interventions to amplify meaningful feature-semantic associations, yielding superior performance on CUB, SUN, and AWA2 datasets compared to state-of-the-art models.

**Strengths:**

The idea  is interesting.

The experiments prove the effectiveness of the method.

**Limitations:**

Some issues need to be addressed:

1. There are some errors in the paper. For example, why is it not Q*K^T in Eq.2?

2. The font in Figure 1 is too small, and the idea is not expressed clearly enough.

3. The author lacks discussion of some related work in the field of zero-shot learning. For example [1].
[1] A causal view of compositional zero-shot recognition

4. Will the code be released publicly?

**Suitability:**

3

---

### Official Review · Reviewer_DWtR · 2024-05-24

**Rating:** 4
**Confidence:** 2

**Summary:**

This paper introduces counterfactual intervention applied to ZSL, and maximize their impacton semantic and target predictions to enhance substantive visual-semantic correlation.

**Strengths:**

Extensive experiments on multiple benchmark datasets demonstrate that the method enhances the substantiveness of
visual-semantic correlation and significantly improves the performance of ZSL.

**Limitations:**

The method's backbone does not correlate well with ViT, so experimental results from other backbones, such as ResNet-101, can be provided to provide more detailed comparisons.

**Suitability:**

3

---

### Official Review · Reviewer_DMUo · 2024-05-25

**Rating:** 2
**Confidence:** 4

**Summary:**

This paper aims to learn substantive visual-semantic correlation and eliminate possible spurious visual-semantic correlation in zero-shot learning. Therefore, this paper proposes to tackle this problem from visual-semantic attention mechanism and causal visual-semantic correlation learning with counterfactual intervention. Experiments show some effectiveness.

**Strengths:**

+This paper introduces causal inference in zero-shot learning to solve visual-semantic correlation issue.

+Experiments show some effectiveness of the proposed method.

**Limitations:**

--There lacks experimental evidence or observation to show the spurious visual-semantic correlation in zero-shot learning.

--There is no visualization comparison before and after removing spurious correlation. It is unclear if the improvement in performance is from the spurious correlation removal.

--The attention mechanism between visual-semantic is not novel because this is common in idea due to the wide use of self- and cross- attention.

--Leveraging counterfactual intervention is widely studied in related fields, such as domain generalization. Therefore, the novelty is limited.

--The writing has a large space for improvement. I obverse a number of typos in language.

--There are some important mathematical symbols missing, such as W_q in Eq.1.

--More ZSL methods are not discussed and compared.

**Suitability:**

2

---

### Official Review · Reviewer_PRM1 · 2024-05-25

**Rating:** 5
**Confidence:** 4

**Summary:**

This paper addresses the challenge of Zero-Shot Learning. Existing methods often directly align visual and semantic features but suffer from mismatched correlations. To overcome this issue, the authors propose a novel method called Causal Visual-Semantic Correlation (CVsC). This method introduces counterfactual intervention to attribute-related visual features, maximizing their impact on semantic and target predictions to reinforce substantive correlations. The proposed method achieves the best results on three benchmark datasets, and ablation studies demonstrate the effectiveness of each component.

**Strengths:**

The paper is well-written and easy to follow.

The motivation for introducing counterfactual intervention in this paper is clear and innovative, and the results are excellent.

**Limitations:**

1 Recent methods in the related work section on Attention Mechanism also need to be discussed, such as [1,2,3]. Similarly, [4] should also be compared and discussed.

[1] MSDN: Mutually Semantic Distillation Network for Zero-Shot Learning. CVPR2022.

[2] Dual Part Discovery Network for Zero-Shot Learning. MM 2022.

[3] DUET: Cross-modal Semantic Grounding for Contrastive Zero-shot Learning. AAAI 2023

[4] Counterfactual Zero-Shot and Open-Set Visual Recognition. CVPR2021

2 There are still questions regarding the design of $\triangle S_1$ and $\triangle S_2$, which require further explanation.

3 From the ablation study, it appears that CCL contributes the most. What could be the possible reasons for this?

**Suitability:**

3

---

### Meta-Review · Area_Chair_wHZ2 · 2024-07-01

**Recommendation:** Accept (Poster)
**Confidence:** 4

**Metareview:**

In this paper, the authors propose the Causal Visual-semantic Correlation method, which uses a Visual Semantic Attention module and Conditional Correlation Loss to learn meaningful visual-semantic correlations, enhanced by counterfactual interventions, resulting in better performance against SOTA methods on three benchmark datasets. Some issues (more discussions on related works, clearer mathematical notations, improved figure representation) need to be addressed. For more details, please refer to the reviewers' detailed comments and final thoughts.